# OpenReview forum: "Uncovering Gaps in How Humans and LLMs Interpret Subjective Language"
_ICLR.cc/2025/Conference — ICLR 2025 Spotlight_

### Official Review · Reviewer_tUBF · 2024-10-17

**Soundness:** 3
**Presentation:** 2
**Contribution:** 3
**Rating:** 6
**Confidence:** 4

**Summary:**

The authors introduced a thesaurus containing pairs of subjective phrases that have dissimilar or similar operational semantics in LLMs. They compare this thesaurus with a human-annotated thesaurus to detect failures in LLMs' understanding (the process of TED). Experimental results under two text generation scenarios show that TED has a high success rate in finding the failure of LLMs' generation.

**Strengths:**

The high-level motivation—finding the gaps between LLMs and humans' understanding, is very important and intriguing. The authors contribute in this direction by uncovering certain phrase pairs where LLMs have an incorrect understanding, causing their generated text to contain unwanted or undesirable properties.

**Weaknesses:**

I think the prominent problem is the writing, which fails to convey the authors' idea clearly. I've read the paper word by word multiple times, but untechnical issues making the paper logically unsmooth and difficult to understand, such as missing clear description of key parts. For example, in Sec 3.2, the process to obtain the operational semantics is missing, instead, it is vaguely described in Sec.4.2. This problem also happens to how the authors construct the semantic thesaurus. Figure 1 is also difficult to understand (see questions).

Another problem is the lack of proof regarding the effectiveness and reliability of their method for obtaining the LLMs' operational semantics $\Delta_w$. The authors also mentioned that they adopted an arbitrary operation.

I also question the usefulness of the proposed LLM operational thesaurus and TED. The phenomenon of LLMs generating unwanted text properties is not new and is often practically addressed by prompting LLMs again to re-generate text while avoiding unwanted properties (a process akin to CoT). This makes the mere presentation of misalignment cases not very useful. The authors also did not discuss how TED can potentially benefit LLMs research and applications in this paper (correct me if I missed). To enhance its practicality, I believe it is good to conduct research and experiment to showcase how to improve LLMs' generation using the proposed TED. However, this paper does not include this part, which diminishes its significance.

**Questions:**

1. What's the definition of *subjective language/phrases*? Can all adjectives be considered as subjective phrases? The authors should clarify this definition at the beginning.
2. In Figure 1, why using orthogonal symbol $\perp$ to denote two antiparallel vectors? Overall, Figure 1 does not present the process of TED clearly. At first glance, Step 1 appears to imply that the thesauruses of LLMs are generated from the text below; in Step 2, it's hard to understand how the judgment of "unexpected side-effect" and "inadquate update" is made. The authors should expand Figrue 1, add more descriptive text on it, and relate to the corresponding sections. Also, the authors is recommended to highlight that Step 2 produces a set of pairs that LLMs misunderstand, which is then evaluated in Step 3.
3. Figure 2 seems not very necessary, as the effect of operational thesaurus vectors is well described in Sec. 3.2. It is better to use the space of Figure 2 to expand Figure 1.
4. In line 147, "A thesaurus describes whether or not phrases are **similar**", similar in what aspect? Please provide more specificity.
5. In Sec. 3.2, it is important to include the process of obtaining $\Delta_w$ here. Regarding to this process which described in line 301-306, which layer's embeddings are used to calculate the gradients? How are the gradients calculated, and why do the gradients can represent operational semantics? The authors should elaborate on this and give theoretical support. Also, have the authors considered the gradients of another tokens? I feel only using the first token can be very biased.
6. In line 207, "we average over gradients from n generic prompts", I am interested in the consistency of the operational semantic vectors of the same word $w$. Could you present the inner-consistency of the vectors (e.g. distribution of their cosine similarity) of the same word $w$ under $n$ different prompts? If the distribution is very divergent, I would highly doubt the effectiveness of the calculated operational semantics.
7. In the following paragraph Building the semantic thesaurus, first, the description of the construction is difficult to understand and the authors are recommended to give examples on it. In line 217, "more aligned" than what? Do you mean if $o_{w_1}$ is expeted to more aligned with $w_2$ than $o_{w_2}$? Besides, I think obtaining human-generated semantic thesaurus may not be that complicated. I expected simply asking humans to indicate weather two phrases have accordant effects (e.g. 'informative' and 'long') or discordant effects (e.g. 'informative' and 'concise'). The authors are recommended to give explaination on why they designed this construction in this manner.
8. In line 232-233, "and less aligned for inadequate updates", I question whether this is a robust metric. For instance, when prompting LLMs to generate longer text such as expanding a research report, we often want them to be just more verbose rather than to add more information (because the imagined information from LLMs can be inaccurate). In this case, when measuring the pair ('informative', 'longer'), it will be identified as 'inadequate update' by the metric, but actuallty but it does not sound inadequate at all. How do you think of this case?
9. In line 274, how exactly do you prompt the LLM? I could not find this information. In line 279, what does "reference subjective phrases" mean? Regarding Inference steering, how exactly do you achieve this?

---

> ### Author Response · Authors · 2024-11-22
> **Response to Reviewer tUBF**
>
> Thank you for your review! We’re glad you thought our high-level motivation is important and intriguing, and appreciate your feedback. Based on the feedback we’ve done the following:
> * Added the experiment you requested where we compare the cosine similarity of the gradients for the same phrase over different prompts; we find that the cosine similarities are routinely high.
> * Added descriptions of how the failures TED uncovers could be resolved with standard methods
> * Added more detailed descriptions to Figure 1, along with additional writing changes.
>
> We’ve also responded to your specific comments below. We hope that if this helps assuage your concerns, you’ll consider raising your score.
>
> ---
>
> _In line 207, "we average over gradients from n generic prompts", I am interested in the consistency of the operational semantic vectors of the same word w. Could you present the inner-consistency of the vectors (e.g. distribution of their cosine similarity) of the same word w under n different prompts? If the distribution is very divergent, I would highly doubt the effectiveness of the calculated operational semantics._
>
> We’ve included a figure with the cosine similarities for different terms in Appendix B.8 (see Figure 5); as you suggest for different phrases, we compute the gradients for many prompts and compute histograms of the pairwise similarities. Overall, we find that the cosine similarities are almost always very high. We also expect there to be some noise in individual examples (since specific outputs to each prompt are stochastic), which is why we average over many gradients to compute embeddings.
>
> However, our empirical results are the main justification of the effectiveness of the calculated operational semantics; as we describe, the only difference between TED and the semantic only baseline is filtering based on our operational semantics. If the operational semantics were not effective, TED would not outperform the semantic only baseline, while in practice it does by large margins
>
> ---
>
> _The authors also did not discuss how TED can potentially benefit LLMs research and applications in this paper (correct me if I missed). To enhance its practicality, I believe it is good to conduct research and experiment to showcase how to improve LLMs' generation using the proposed TED. However, this paper does not include this part, which diminishes its significance._
>
> Thanks for this suggestion; in the original draft, we discuss one potential method for improving models (training the LLM so its operational thesaurus matches the reference), in lines 525-531. Based on your suggestion, we’ve also added more actionable guidelines on how TED might improve systems; these include helping practitioners choose better system prompts, curating fine-tuning sets based on them, or using the human thesaurus as a direct supervision signal (see response to all reviewers). We think both adjusting the prompt and fine-tuning could be implemented easily with standard methods; we view the primary challenge and contribution of our work as finding these failures out of the large space of possibilities.
>
> ---
>
> _The authors should expand Figrue 1, add more descriptive text on it, and relate to the corresponding sections. Also, the authors is recommended to highlight that Step 2 produces a set of pairs that LLMs misunderstand, which is then evaluated in Step 3_
>
> Thanks for pointing this out; we’ve adjusted this figure in the following ways:
> * We change orthogonal and perpendicular to similar (SIM) and dissimilar (DIS)
> * We changed the title of Step 2 to “Identify Failures (Clashes)” to make it clear these are things the LLMs misunderstands
> * We include a more clear arrow from the pair in Step 2 to the output in Step 3 to make it more clear that the failures come from Step 2. We additionally made the label of Step 3 "Test Failures downstream" so it more clearly links from Step 2.
> * We made the caption more descriptive, and added examples
>
> We hope this makes it easier to parse, and helps resolve your concerns.
>
> ---
>
> _I think the prominent problem is the writing, which fails to convey the authors' idea clearly. I've read the paper word by word multiple times, but untechnical issues making the paper logically unsmooth and difficult to understand, such as missing clear description of key parts._
>
> We respond to many of the specific writing points you raise below, but one high-level design choice that we think caused confusion is our decomposition of Section 3 and Section 4; in Section 3, we describe TED and ways to evaluating TED in the abstract, then include our specific instantiation (with many of the details you asked for in Section 3) in Section 4. We did this because TED is general, and we expect it to work for many choices of prompt, embedding, and judge. In the revision, we forward-reference that we decompose in this way at the beginning of Section 3 to make this more explicit.
>
> ---
>
> [response continued below]

---

> > ### Author Response · Authors · 2024-11-22
> > **Response (continued)**
> >
> > [continued]
> >
> > ---
> >
> > _I also question the usefulness of the proposed LLM operational thesaurus and TED. The phenomenon of LLMs generating unwanted text properties is not new and is often practically addressed by prompting LLMs again_
> >
> > TED does more than flag unwanted text properties of specific outputs; we classes of many inputs on which the LLM acts in a particular way (outputs are more like some subjective phrase than what humans expect) systematically; our results find many failures that occur on over 90% of prompts that include the subjective phrase (e.g., 57% of unexpected side-effects for Llama 3 8B). Because these are targeted and systematic, we expect them to be easy to intervene on (see next response), and are not just properties of specific outputs.
> >
> > ---
> >
> > _What's the definition of subjective language/phrases? Can all adjectives be considered as subjective phrases? The authors should clarify this definition at the beginning._
> >
> > We added a clarification in the main body; subjective phrases includes any language that can be systematically added to prompts to steer LLMs (including adjectives, few word phrases, or even entire instructions). TED seamlessly handles multi-token phrases since it elicits information about the phrase through changes in output.
> >
> > ---
> >
> > _In line 147, "A thesaurus describes whether or not phrases are similar", similar in what aspect? Please provide more specificity._
> >
> > Similar in terms of operational semantics; we describe this in lines 152-157 of the original draft.
> >
> > ---
> >
> > _In Sec. 3.2, it is important to include the process of obtaining Δw here. Regarding to this process which described in line 301-306, which layer's embeddings are used to calculate the gradients? How are the gradients calculated, and why do the gradients can represent operational semantics?._
> >
> > We give the theoretical construction in Equation 1, and describe the implementation details in the “LLM operational thesaurus section of 4.2. We take gradients via backprop on the log loss we describe in Equation 1; we do this using a single forward and backward pass per prompt.
> >
> > ---
> >
> > _In line 274, how exactly do you prompt the LLM? I could not find this information_
> >
> > Sorry about this; we included the prompt in the original version in Appendix B.2.2, and now link to it in the revision.
> >
> > ---
> >
> > _In the following paragraph Building the semantic thesaurus, first, [...] In line 217, "more aligned" than what? Do you mean if  o(w1)  is expeted to more aligned with w2 than o(w2)?_
> >
> > The judge evaluates which of o(w1) and o(\empty) is more like w2 (e.g., is more “informative”) as we describe in lines 231-234 of the original paper. We do this to isolate the effect of adding the phrase w1; we solicit labels from humans to evaluate whether including w1 will steer the output towards w2, and only flag failures when including w1 steers the output in a way that the human annotators do not expect.
> >
> > ---
> >
> > _Besides, I think obtaining human-generated semantic thesaurus may not be that complicated. I expected simply asking humans to indicate weather two phrases have accordant effects (e.g. 'informative' and 'long') or discordant effects (e.g. 'informative' and 'concise')_
> >
> > We describe our instantiation of the human semantic thesaurus in lines 310 - 316 and include the exact prompt in Appendix B.7; the prompt is similar to what you suggest.
> >
> > ---
> >
> > _In line 232-233, "and less aligned for inadequate updates", I question whether this is a robust metric. For instance, when prompting LLMs to generate longer text such as expanding a research report, we often want them to be just more verbose rather than to add more information (because the imagined information from LLMs can be inaccurate). [...] How do you think of this case?_
> >
> > Our failures are definite with respect to actual human annotators; if the annotators also think that making a research report longer shouldn’t necessarily make it more informative (e.g., they label as “unsure”), this will not be counted as an inadequate update.
> >
> > ---
> >
> > _In line 274, how exactly do you prompt the LLM? I could not find this information. In line 279, what does "reference subjective phrases" mean? Regarding Inference steering, how exactly do you achieve this?_
> >
> > We include the specific prompts used in Appendices B.2.2 and and B.5 of the original version. Thanks for flagging this; we forgot to link these in S4.1 (we instead included examples rather than the template in the main body); we’ve included links in the revised version.
> >
> >
> > ---
> >
> > _Another problem is the lack of proof regarding the effectiveness and reliability of their method for obtaining the LLMs' operational semantics ._
> >
> > We think TED works since our embeddings capture how a latent LLM embedding would need to change to mimic the effect of adding the subjective phrase to the prompt  [see 196-200]. Nevertheless, the primary justification of our method is its empirical success in finding failures.
> >
> > ---
> >
> > Please let us know if you have additional questions!

---

> > > ### Comment · Reviewer_tUBF · 2024-11-22
> > > **Reply From Reviewer**
> > >
> > > Thanks the authors for answering my questions.
> > >
> > > I have reviewed their responses and the revision. They have made the necessary improvements and edits to the submission draft to enhance clarity. Additionally, they conducted experiments on the consistency of the operational semantic vectors, and the results align with their findings.
> > >
> > > I have increased my rating.

---

### Official Review · Reviewer_cGyi · 2024-11-04

**Soundness:** 3
**Presentation:** 3
**Contribution:** 4
**Rating:** 8
**Confidence:** 4

**Summary:**

This study introduces a novel method to identify misalignments in how LLMs interpret subjective prompts compared to human expectations. TED constructs an operational thesaurus based on the LLM’s interpretations of phrases like “enthusiastic” or “witty” and compares it to a human-created semantic thesaurus. Misalignments are flagged as “unexpected side effects” or “inadequate updates,” depending on whether the model's output behavior deviates from or fails to meet human expectations.

The authors evaluated TED on two tasks—output editing and inference steering—demonstrating its effectiveness in uncovering surprising behaviors. For instance, TED detected that prompting models for “enthusiastic” outputs sometimes resulted in “dishonest” outputs. This method highlights TED’s value in addressing alignment challenges in subjective language, proposing it as a scalable tool to enhance the reliability of LLMs in aligning with human intent.

**Strengths:**

Originality:
* The paper introduces TED, a unique approach to identifying LLM misinterpretations in subjective language by constructing and comparing operational and semantic thesauruses. This structured focus on “unexpected side effects” and “inadequate updates” adds a fresh, nuanced perspective to alignment research, addressing a critical yet underexplored aspect of model behavior.
Quality:
* TED’s methodology is rigorous, with careful construction of thesauruses using embedding and gradient-based techniques. Quantitative results demonstrate TED’s effectiveness over baseline methods, and the authors provide a balanced view by discussing limitations, lending credibility and depth to their findings.

Clarity:
* The paper is well-organized and accessible, effectively explaining complex methods and illustrating key points with clear examples (e.g., “enthusiastic” prompting unintended “dishonest” outputs). The writing maintains a technical depth while remaining readable, supporting a broad audience’s understanding.

Significance:
* The work addresses a crucial challenge: aligning LLM interpretations with human expectations in subjective contexts. TED’s contributions extend alignment research to emotional and tonal aspects, impacting user-centered LLM applications and offering a scalable approach for early misalignment detection in model development.

**Weaknesses:**

Context-Sensitivity in Embedding Representation:
* The paper mentions embedding each phrase independently of context, which may overlook nuances in interpretation that depend on the type of task (e.g., writing a "witty blog" versus a "witty proposal"). Incorporating context-dependent embeddings could enhance TED’s robustness by tailoring thesaurus creation based on usage scenarios. This could be achieved by developing context-specific operational thesauruses or dynamically updating embeddings based on task context, potentially using attention-based methods to focus on relevant contextual tokens.

Impact of GPT-4 Judgments on Validation:
* The paper relies on GPT-4 for validating downstream failures, but this could introduce bias since GPT-4 is itself an LLM with its own alignment characteristics. Exploring alternative or supplementary validation approaches, such as using human expert reviews or a consensus-based scoring system from multiple models, could mitigate this potential bias. Additionally, assessing whether GPT-4's validation aligns with human judgments on subtle or ambiguous failures would reinforce the robustness of the evaluation.

**Questions:**

Q1: How does TED handle potential context-specific interpretations of subjective phrases (e.g., “witty” in different tasks like blogs vs. proposals)?

Q2: Given the reliance on GPT-4 to validate TED’s flagged misalignments, how do you address potential alignment biases that GPT-4 might introduce?

Q3: Did you observe specific hierarchies or dependencies between phrases (e.g., “intelligent” often implying “engaging”)? If so, how did TED handle these dependencies?

Q4: Beyond identifying misalignments, have you considered how TED’s findings might guide improvements in LLM training or alignment processes?

---

> ### Author Response · Authors · 2024-11-22
> **Response to Reviewer cGyi**
>
> Thank you for your review of our work! We’re glad you found it offers a “fresh, nuanced perspective to alignment research”, that we “address a crucial challenge”, and our “methodology is rigorous”. We respond to your questions and concerns below.
>
> ---
>
> _The paper relies on GPT-4 for validating downstream failures, but this could introduce bias since GPT-4 is itself an LLM with its own alignment characteristics. Exploring alternative or supplementary validation approaches, such as using human expert reviews or a consensus-based scoring system from multiple models, could mitigate this potential bias.  Given the reliance on GPT-4 to validate TED’s flagged misalignments, how do you address potential alignment biases that GPT-4 might introduce?_
>
> Thanks for raising this; we agree, and we’ve added human validation on one part of the pipeline that doesn’t currently have human supervision (supervising GPT-4 on the judge over outputs). We find that humans tend to agree with LLMs, and that if anything LLMs underestimate the rate at which TED produces failures; see details in the response to all reviewers and Appendix B.7.3 of the revised paper.
>
> ---
>
> _The paper mentions embedding each phrase independently of context, which may overlook nuances in interpretation that depend on the type of task (e.g., writing a "witty blog" versus a "witty proposal"). Incorporating context-dependent embeddings could enhance TED’s robustness by tailoring thesaurus creation based on usage scenarios. This could be achieved by developing context-specific operational thesauruses or dynamically updating embeddings based on task context, potentially using attention-based methods to focus on relevant contextual tokens._
>
> This is a good point; we discuss this in lines 474-481 of our submission. One benefit of TED is that it puts no constraints on what counts as a phrase, so it can seamlessly extend to these settings.
>
> ---
>
> _How does TED handle potential context-specific interpretations of subjective phrases (e.g., “witty” in different tasks like blogs vs. proposals)?_
>
> Right now TED does not handle context, although TED could be seamlessly extended to do so (by only averaging gradients where a term is used in a specific context). We think this is an exciting direction for subsequent work.
>
> ---
>
> _Did you observe specific hierarchies or dependencies between phrases (e.g., “intelligent” often implying “engaging”)? If so, how did TED handle these dependencies?_
>
> Right now TED doesn’t handles these dependencies; it’s currently symmetric. We expect that there are ways to extend TED to handle them down the line; for example, we could measure the gradient of “engaging” outputs relative to “intelligent” prompts and vice versa; if “intelligent” implies “engaging” but not the reverse, the gradients should be large on “engaging” outputs but not on “intelligent” outputs. We think adding this as a post-hoc step is an exciting direction for subsequent work.
>
> ---
>
> _Beyond identifying misalignments, have you considered how TED’s findings might guide improvements in LLM training or alignment processes?_
>
> Great question; in the original draft, we discuss one potential method for improving models (training the LLM so its operational thesaurus matches the reference), in lines 525-531. We’ve also added more actionable guidelines on how TED might improve systems; these include helping practitioners choose better system prompts, curating fine-tuning sets based on the , or supervising on the internal thesaurus directly (see response to all reviewers). We think both adjusting the prompt and fine-tuning could be implemented easily with standard methods; we view the primary challenge and contribution of our work as finding these failures out of the large space of possibilities.
>
> ---
>
> Please let us know if you have any additional questions!

---

### Official Review · Reviewer_WNxc · 2024-11-06

**Soundness:** 3
**Presentation:** 2
**Contribution:** 2
**Rating:** 8
**Confidence:** 3

**Summary:**

The paper introduces a novel approach, TED (Thesaurus Error Detector), to detect misalignment between language models' interpretation of subjective prompts and human expectations. Using TED, the authors develop an operational thesaurus for subjective phrases, comparing it with human-constructed semantic thesauruses to identify discrepancies. TED aims to uncover unexpected behavior in LLMs by examining how models handle subjective instructions, such as "enthusiastic".

**Strengths:**

1. Novel Methodology: TED presents an interesting approach to detect model misalignments with human expectations, filling a critical gap in aligning LLM behavior with human intent.

2. Significance: Subjective language interpretation is an important yet challenging task nowadays.

**Weaknesses:**

1. Lack of Practical Recommendations: While TED detects misalignments effectively, the paper could provide more actionable guidelines or solutions to mitigate these issues in real-world applications.

2. Ambiguity in Evaluation Metrics: The evaluation metrics for TED’s effectiveness could be clarified, as the success rates reported could benefit from further contextualization to understand their practical implications. Also, it would be better to clarify how exactly the success rate is calculated.

3. Need more justification of the results: Through empirical tests, TED reportedly achieves higher success rates in detecting inadequate updates than a semantic-only baseline, while both approaches yield fewer failures overall. However, the implications of these results need clearer justification and relevance to real-world LLM applications.
For instance, the statement "TED additionally finds inadequate updates with higher success rates than the semantic-only baseline, but both TED and the baseline find fewer failures overall" is an interesting finding, but it’s unclear what this means in practical terms.

**Questions:**

In line 283, "For example, the LLM might language model to write a “witty” essay or an “accessible” blogpost about machine learning."

What does "the LLM might language model" mean?

---

> ### Author Response · Authors · 2024-11-22
> **Response to Reviewer WNxc**
>
> Thanks for your review of our work! We’re glad you thought our approach is “interesting” and is “filling a critical gap in aligning LLM behavior with human intent”. We respond to your questions and comments below.
>
> ---
>
> _While TED detects misalignments effectively, the paper could provide more actionable guidelines or solutions to mitigate these issues in real-world applications._
>
> Thanks for this suggestion; in response we’ve also added more actionable guidelines on how TED might improve systems; these include helping practitioners choose better system prompts, curating fine-tuning sets based on them, or using the human thesaurus as a direct supervision signal (see response to all reviewers). We think both adjusting the prompt and fine-tuning could be implemented easily with standard methods; we view the primary challenge and contribution of our work as finding these failures out of the large space of possible behaviors.
>
> ---
>
> _The evaluation metrics for TED’s effectiveness could be clarified, as the success rates reported could benefit from further contextualization to understand their practical implications. Also, it would be better to clarify how exactly the success rate is calculated._
>
> We describe the calculation of the success rate in lines 231-234 of the original paper, and the actual implementation details we use in the _Evaluating TED_ paragraph of section 4.2. Intuitively, the success rate for a pair of phrases describe what fraction of the time (on held out prompts) including the phrase produces the unintended consequence or the inadequate updates; this aims to capture how frequently the failure arises at deployment. Both high and low success rates have important practical ramifications; a low success rate (e.g., 0.1) could still be too high for failures like producing “dishonest” outputs when humans prompt for enthusiasm. The aggregate success rates, described between lines 235 - 240 and reported in the Figures, primarily capture TED’s overall efficacy.
>
> ---
>
> _However, the implications of these results need clearer justification and relevance to real-world LLM applications. For instance, the statement "TED additionally finds inadequate updates with higher success rates than the semantic-only baseline, but both TED and the baseline find fewer failures overall" is an interesting finding, but it’s unclear what this means in practical terms._
>
> The main observation in this statement is that inadequate updates seem to occur less frequently (or are at least harder to find for both the semantic only baseline and TED) than unexpected updates; in the original version, we say “[this] indicates that inadequate updates are less frequent in practice than unexpected side effects.” We might intuitively expect this since there are more ways of adding unexpected attributes than missing necessary attributes. This also practically means that developers should actively test for a wide range of unexpected side-effects.
>
> ---
>
> _In line 283, "For example, the LLM might language model to write a “witty” essay or an “accessible” blogpost about machine learning." What does "the LLM might language model" mean?_
>
> Thanks for flagging this; we mean a “user might prompt an LLM to write…”. We updated this in the revision.
>
> ---
>
> Please let us know if you have any additional questions!

---

> > ### Comment · Reviewer_WNxc · 2024-11-24
> > **Thank the authors for their clarification.**
> >
> > Thanks the authors for their clarification! After reviewing the authors responses and other reviewers' review, I increased my score.

---

### Official Review · Reviewer_pBCH · 2024-11-09

**Soundness:** 3
**Presentation:** 3
**Contribution:** 3
**Rating:** 8
**Confidence:** 3

**Summary:**

This paper identifies misalignment between human intent and LLM behavior adjustments (operational semantics) when prompted with subjective language.

They introduce a method called thesaurus error detector (TED). The method first involves constructing a similarity matrix comparing LLMs’ operational semantics for different subjective phrases (approximated by embedding gradients), and then elicits failures by identifying how this thesaurus disagrees with with a reference thesaurus (created based on either human or stronger LLM annotations).

The authors’ experiments show that TED consistently outperforms a semantic-only baseline for two models (Mistral 7B and Llama 3 8B) in identifying two kinds of errors (unexpected side effects and inadequate updates) for two kinds of tasks (output editing and inference steering).

The authors also present some qualitative examples throughout the paper, such as Mistral 7B producing more “harassing” outputs when instructed to make articles “witty”.

**Strengths:**

- The paper is well-written and experimental design choices are remarkably well-documented, which facilitates reproducibility experiments and future work to easily extend this work to different settings.

- The thesaurus based method is interesting and creative

- I find the author’s broader contribution of “characterizing LLM behavior via relationships between abstract concepts rather than individual outputs directly” to be very compelling

**Weaknesses:**

- Given that this work is ultimately about *meaning* and the word *semantics* is used throughout the paper, there is surprisingly no connection to any relevant concepts or prior literature in subjectivity, semantics, or pragmatics. This hinders the work’s conceptual clarity and contribution.

- The authors rely on GPT-4 in many parts of the pipeline: identifying subjective phrases, constructing the reference thesaurus, and acting as a judge. These decisions are well-motivated but some human validation is needed at each stage, perhaps on just a small sub-sample. Especially because this work focuses on biases and limitations of LLMs, it’s hard for me to switch to fully trusting GPT-4’s outputs.

- For the human evaluation, there should be some sort of inter-annotator agreement recorded (again, maybe on a small subset). Because we’re dealing with subjectivity, humans’ interpretations of each phrases and the relationship between phrases would likely vary widely.

- One area that remains unclear to me is in the motivation for testing TED on LLM responses to this particular kind of ethical question. The examples in the prompt all start with “why is it okay”, which implies that “it is okay” which may bias both LLM outputs and the constructed thesaurus.

**Questions:**

- How do you define subjectivity?

- And given that subjectivity leads to different human interpretations, what do we want from LLMs?

- More specifically, I'm not convinced based on the provided examples that unexpected side effects necessarily indicate model failures, as suggested by the language throughout the paper. I can't easily imagine what it would mean for subjective phrases to affect LLM outputs along just that one dimension (enthusiastic, witty, etc.) _without_ affecting other dimensions?

- Why don’t you use an actual thesaurus or similar lexical resources such as WordNet or ConceptNet as the reference?

- Relatedly, why not use existing subjectivity lexicons rather than rely on GPT-4 to generate the set of phrases for this work? While there is manual curation mentioned in the appendix, there may be recall biases in kinds of subjective phrases that GPT-4 simply doesn’t surface.

- I’d recommend connecting this work to ideas in semantics and pragmatics that focus on the relationships between statements: entailment/natural language inference, presuppositions, and implicature. I think this could help ground the method, annotation, and evaluation.

- How should I think about the selection of the embedding to calculate the gradient with respect to? Why do you expect any other token or internal activation to yield similar results?

---

> ### Author Response · Authors · 2024-11-22
> **Response to Reviewer pBCH**
>
> Thank you for your thoughtful review of our work! We’re glad you thought our method is “interesting and creative”, and that our contribution of “characterizing LLM behavior via relationships” to be “very compelling”. We respond to your comments below:
>
> ---
>
> _The authors rely on GPT-4 in many parts of the pipeline: identifying subjective phrases, constructing the reference thesaurus, and acting as a judge. These decisions are well-motivated but some human validation is needed at each stage, perhaps on just a small sub-sample. Especially because this work focuses on biases and limitations of LLMs, it’s hard for me to switch to fully trusting GPT-4’s outputs._
>
> Thanks for raising this; we agree, and we’ve added human validation on one part of the pipeline that doesn’t currently have human supervision (supervising GPT-4 on the judge over outputs). We find that humans tend to agree with LLMs, and that if anything LLMs underestimate the rate at which TED produces failures; see details in the response to all reviewers and Appendix B.7.3 of the revised paper.
>
> ---
>
> _For the human evaluation, there should be some sort of inter-annotator agreement recorded (again, maybe on a small subset). Because we’re dealing with subjectivity, humans’ interpretations of each phrases and the relationship between phrases would likely vary widely._
>
> This is a good point; in our study we have three annotators label each pair of phrases, and only include pairs where all three annotators agree (see 315-316). We also report aggregate statistics about agreement in Figure 4; the annotators all agreed on around 60% of pairs (whereas they only would have agreed on 11% via random chance).
>
> ---
>
> _Given that this work is ultimately about meaning and the word semantics is used throughout the paper, there is surprisingly no connection to any relevant concepts or prior literature in subjectivity, semantics, or pragmatics. [...] I’d recommend connecting this work to ideas in semantics and pragmatics that focus on the relationships between statements: entailment/natural language inference, presuppositions, and implicature. I think this could help ground the method, annotation, and evaluation._
>
>
> Thanks for raising this! We definitely think there are close relationships to concepts from the prior literature, and added a paragraph in related work to describe this. We think our work closely ties with pragmatic implication; it’s possible that some of the failures come from LLMs imitating humans’ pragmatic implications of phrases (e.g., it’s possible that humans actually tend to say more hateful things when asked to be witty.)
>
> That being said, TED’s only goal is to uncover scenarios when using subjective phrases in the prompt will produce outputs that are different from what prompters expect, without attributing cause. Our evaluation directly tests for this: we ask human annotators (that simulate prompters) whether or not they expect outputs to have certain properties, and view any deviation from these expectations as failures. These failures will become especially salient as models are prompted by humans, but deployed with limited oversight.
>
> ---
>
> _One area that remains unclear to me is in the motivation for testing TED on LLM responses to this particular kind of ethical question. The examples in the prompt all start with “why is it okay”, which implies that “it is okay” which may bias both LLM outputs and the constructed thesaurus._
>
> This is a great question; the ethical questions come up in the output editing setting that is designed to mimic Constitutional AI. In this setting, the goal is to get the model to adhere to a constitution, especially when responding to challenging topics with potential bias. However, TED finds failures in the inference steering setting too, where we use much more standard prompts (e.g., “Write an enthusiastic letter to a school board proposing the introduction of coding classes in middle schools”), so this is not a critical part of the setup.
>
> ---
>
> _More specifically, I'm not convinced based on the provided examples that unexpected side effects necessarily indicate model failures, as suggested by the language throughout the paper. I can't easily imagine what it would mean for subjective phrases to affect LLM outputs along just that one dimension (enthusiastic, witty, etc.) without affecting other dimensions?_
>
> To clarify, we say that a failure occurs when outputs are affected along directions that human annotators don’t expect, rather than all directions; this means that most effects are not considered failures (e.g., prompting for “enthusiastic” and getting “energetic” or “long” would not be failures). When collecting e.g., unexpected updates, we allow the annotator to specify whether or not they expect that prompting a model with one phrase affects the output along another axis, and only count failures that go against all annotators expectations.

---

> > ### Author Response · Authors · 2024-11-22
> > **Response (continued)**
> >
> > [continued from above]
> >
> > ---
> >
> > _Why don’t you use an actual thesaurus or similar lexical resources such as WordNet or ConceptNet as the reference?_
> >
> > We’re especially worried about misalignment with respect to what a user expects and what the language model actually does, so we aimed to directly emulate users. Actual thesauruses tend to have false-positives in this setting (for example, thesaurus.com says detailed is synonymous with “complicated” and “accurate”, but these should intuitively produce very different language model responses). Nevertheless, we think coming up with better thesauruses (by combining humans with existing references and better aggregating feedback from multiple humans) is an important direction for subsequent work.
> >
> > ---
> >
> > _Relatedly, why not use existing subjectivity lexicons rather than rely on GPT-4 to generate the set of phrases for this work? While there is manual curation mentioned in the appendix, there may be recall biases in kinds of subjective phrases that GPT-4 simply doesn’t surface._
> >
> > Good question; similar to the above, we especially wanted phrases that people would actually use interacting with language models, and methods that didn’t use GPT-4 have lots of false positives (e.g., include phrases that users wouldn’t likely query a language model with, and thus wouldn’t be interesting failures). That being said, we already combine GPT-4 with other references (such as taking the adjectives from Claude’s constitution), and think that including salient terms that GPT-4 misses would only increase the set of failures that TED exposes.
> >
> > ---
> >
> > _How should I think about the selection of the embedding to calculate the gradient with respect to? Why do you expect any other token or internal activation to yield similar results?_
> >
> > We want some embedding that causally impacts the language model output in order to simulate adding in the subjective phrase; these embeddings let us compare whether or not the causal interventions for terms are similar or different (for example, if two phrases are truly synonyms to the model, the changes in embedding will be the same). We expect that the relative comparisons between embeddings (especially for very similar or very different embeddings) would be similar across many tokens or internal activations, even though actual embeddings are different.
> >
> > ---
> >
> > Please let us know if you have any other questions!

---

> > > ### Comment · Reviewer_pBCH · 2024-11-23
> > > **Thanks!**
> > >
> > > Thank you for the thorough response and clarifications. I have adjusted my score accordingly.

---

### Author Response · Authors · 2024-11-22
**Response to all reviewers**

We thank all reviewers for their thoughtful feedback on our work. Reviewers found our goal of finding gaps between the LLMs and human’s understanding _“very important and intriguing”_ (tUBF), characterized our work as a _“fresh, nuanced perspective”_ to alignment research (cGyi) that _“fill[s] a a critical gap in aligning LLM behavior with human intent”_ (WNxc). Reviewers also found our thesaurus method _“interesting and creative”_ (pBCH) and our methodology _“rigorous”_ (cGyi). Most reviewers additionally appreciated the quality of our writing; saying that the _“paper is well-written and experimental design choices are remarkably well-documented, which facilitates reproducibility experiments and future work”_ (pBCH) and the paper is _“well-organized and accessible”_ (cGyi)

Multiple reviewers wanted justification of the use of GPT-4 as a judge of outputs, and practical recommendations for how TED could be used to improve models. To address these, we’ve included an additional human study validating the use of GPT-4 as a judge, and added discussion of practical recommendations to the paper. We elaborate on both of these below, and answer other questions in the response to individual reviewers.

---

_Human validation for the use of GPT-4 as a judge._

Several reviewers also hoped for some validation that the success of TED isn’t due to the GPT-4 judge on outputs. To test this, we tried using humans to judge a subset of outputs; in particular, we take 200 out of the 24,000 total comparisons from the paper, and label each with three humans from Mechanical Turk. ___We found that the aggregate humans largely matched the LLM’s judgment; in particular:
* Of the 200 examples, the majority-voted annotator judgment (the output that either 2 or 3 of the three annotators choose) matched the language model 84% of the time.
* Each individual annotator only matched the majority annotator judgment (composed of them and two other annotators) 91% of the time. This means that the LLM is nearly as good at predicting the majority of annotators as each individual annotator (and would likely get closer with more annotators, since each annotator would be less correlated with the majority)
* On outputs where all three annotators agree (74% of prompts), the language model agrees with the annotators 97% of the time
* Humans tended to say the subjective output was a failure _more frequently_ than the LLM; the LLM says TED’s failures are successful 86% of the time, compared to 97% for the majority annotator judgment

Overall, this indicates that the LLM annotations are similar to humans, and the LLM might actually be _underestimating_ TED’s efficacy. We include the specific data collection pipeline and additional details in Appendix B.7.3. This study means __we have now included human validation at each point in the pipeline__; both for constructing thesauruses, and validating that failures arise downstream.

---

_Practical recommendations for using TED to improve systems_

Multiple reviewers asked for practical recommendations about how TED could be used to improve models. We’ve expanded on our discussion section describing this in the revision, but to summarize there are three main recommendations:
* TED could help system developers design system prompts; system prompts frequently use subjective language, and TED could help developers choose language with fewer negative side-effects. For example, instead of prompting the model to give “enthusiastic” responses to questions about health (and thus get “dishonest” outputs), a prompter could swap in “energetic” to avoid dishonesty.
* TED allows for targeted fixes to models directly via fine-tuning; given a failure that TED finds (prompting models to be “enthusiastic” makes them “dishonest”), developers could collect data without the failure (“enthusiastic” outputs that are “honest”), then fine-tune the model on this data using SFT or DPO. This is intractable to do for all of the (quadratically many) failures, but TED identifies the much smaller set of failures the LLM actually exhibits
* In the future, TED’s thesauruses could provide direct supervision signal; as we describe in lines 525-531 of our original submission, we could supervise language models to match human thesauruses directly.

We think the first two can be implemented today by applying standard prompting or fine-tuning strategies, and the third is a promising line for subsequent work.

---

### Meta-Review · Area_Chair_KfAj · 2024-12-19

**Metareview:**

**Summary:**

The authors propose a method (TED) for detecting errors in AI model alignment where subjective language is defined by LM's operational semantics in a way that deviates from human intentions. Using LLM deployment settings of editing text to adhere to a subjective characteristic (output-editing) or generating new text that adheres to a characteristic (inference-steering), they compare similarity between subjective phrases based on their influence over a LLM's output (an operational thesaurus) to the intended similarity as annotated by human labelers (a semantic thesaurus). Their findings uncover noteworthy and harmful cases of misalignment between LM definitions and human intentions, for example Llama 3 associating "enthusiasm" with "dishonesty."

**Strengths:**

- Creative, clever methodology for developing a LLM thesaurus that is directly comparable to human concepts of semantic similarity

- The findings from the paper and TED framework could lead to progress in humanlike automated interpretation of subjective language, a longstanding challenge in NLP research

**Weaknesses:**

- My primary concern was the use of GPT-4 validation, but this has been suitably addressed in the authors' rebuttal with human/GPT-4 comparison. A limitation of the work, acknowledged by the authors in their rebuttal, is the context-free nature of the subjective phrase representations. However, this is necessary to isolate internal definitions and I agree context-awareness is out of scope. It seems like the work would be easily extended to a more context-aware setting.

I believe this is one of the most interesting AI alignment papers I have seen recently, and I agree with the reviewers that it is a fresh approach on quantifying differences between human and LLM interpretations of language. I recommend acceptance.

**Additional Comments On Reviewer Discussion:**

The reviewers are unanimously supporting acceptance, and I see no reason to disagree with them. The work is well-motivated, very well-written after the authors' clarifications in the rebuttal, novel, and presents a simple method that could have a meaningful impact on improving human-AI alignment. The primary concerns raised were (1) lack of connection to linguistics literature on subjectivity/pragmatics, (2) use of GPT-4 as a judge without human validation,  (3) lack of discussion around real-world applications, and (4) validity of their gradient-based approach for obtaining LLM operational semantics (e.g consistency of gradients that are averaged to construct context-free subjective phrase embedding vectors). According to the reviewers and my own analysis, they have satisfactorily addressed these issues in the revised version of the paper. The method is effective for uncovering errors where subjective characteristics are inadequately represented or incorrectly associated.

Many points raised by the reviewers could open up future work I'd encourage the authors to pursue: for example, the authors' choice of only including subjective phrase pairs with agreement among 3 human annotators is reasonable, but I'm curious what the results would be for more divisive pairings. I am also curious, especially given recent literature around sociodemographic effects on annotator disagreement, whether annotator demographics or cultural background could affect results. Lastly, as I state above, I agree with reviewer cGyi that context-awareness would be worth exploring in future work.

---

### Decision · Program_Chairs · 2025-01-22

Accept (Spotlight)